# Diagnostic and Therapeutic Evaluations of Computed Tomography in Three Calves with Unilateral Otitis Media Treated with Ventral Bulla Osteotomy

**DOI:** 10.3390/vetsci9050218

**Published:** 2022-04-29

**Authors:** Takeshi Tsuka, Takao Amaha, Yoshiharu Okamoto

**Affiliations:** Clinical Veterinary Sciences, Joint Department of Veterinary Medicine, Faculty of Agriculture, Tottori University, 4-101, Koyama-Minami, Tottori 680-8550, Japan; amaha@tottori-u.ac.jp (T.A.); yokamoto@tottori-u.ac.jp (Y.O.)

**Keywords:** calf, computed tomography, otitis media, ventral bulla osteotomy

## Abstract

This case report describes the clinical utility of computed tomography (CT) in preoperative evaluation of otitis media in three calves, each exhibiting unilateral head tilt and ear droop. Of the three animals examined by CT, right-sided and left-sided involvements of this disease could be diagnosed in two animals and one animal, respectively, as represented by the accumulations of hyperattenuating contents within the extended tympanic bulla. These cases were treated with a ventral bulla osteotomy, which was conducted between the sternothyroid muscle and the omohyoid muscle via an incision made caudally to the area of the thyroid cartilage. The postoperative outcome was satisfactory in one case. However, the remaining two cases showed unsatisfactory outcomes; one calf died on the 10th postoperative day, as it was already seriously ill. The other animal died suddenly during surgery, potentially due to stimulation of the vagus nerve. The use of CT could allow effective visualization of unilateral lesions, proving helpful for the recommendation of surgical intervention. Based on the unfavorable results in two cases, we note that to prevent surgical complications, the clinical applicability of CT such as the creation of severity criteria and development of imaging-assist procedure should be advanced.

## 1. Introduction

Otitis media is one of the most common infectious diseases in younger calves [1,2,3]. The diagnosis of bovine otitis media is commonly obtained through specific clinical signal appearance and laboratory tests such as isolation of pathogenic microorganisms from deep nasal pharyngeal swabs and bronchial alveolar lavage [3,4]. Radiography is the traditional common imaging modality used to observe otitis media, although it provides inferior image quality where the structures of the ear and tympanic bulla are both difficult to distinguish [2,3,4]. Additionally, the radiographic technique taken obliquely at 20° is required for visualizing each structure of the tympanic bulla, which prevents overlapping between targeted tympanic bulla and the peripheral osseous structures or opposite tympanic bulla [2,4,5]. More recently, computed tomography (CT) has been touted as the gold standard ante mortem diagnostic imaging tool for bovine otitis media, providing a simultaneous evaluation of both the accumulation of exudates into the lumen of the tympanic bulla, and the osseous changes in tympanic bulla, such as extension, osteolysis, and osteoproliferation on the same section [2,4]. Thus, CT is superior to radiography in its diagnostic sensitivity and specificity [4,5]. However, few bovine reports have previously described the utility of CT in preoperative evaluation [6,7].

Continuous, long-term administration of antibiotics and anti-inflammatory drugs is the first choice of treatment for bovine otitis media [1,6,7,8,9]. Irrigation into the affected tympanic cavity is chosen as an advanced conservative therapy [1,10]. The irrigation procedure—myringotomy—is commonly achieved via a perforation made in the tympanic membrane using a sharp metal needle inserted into the ear canal [1]. As part of a modified myringotomy, an indwelling needle intravenous cannula has been previously utilized to perforate the tympanic membrane for subsequent irrigation, with pressure generated by a syringe [11]. Some practitioners have performed high-pressure irrigation into the tympanic cavity using a continuous drench gun, with an oral jet irrigator modified from the human otologic therapy, obtaining the simultaneous effects of myringotomy and intra-ear irrigation [11,12,13,14]. High-pressure irrigation using a continuous drench gun could contribute to a higher healing rate (73.3%) than that of myringotomy (25.0%) [12]. Conversely, high-pressure irrigation utilizing a modified myringotomy allowed for a higher healing rate than the drench gun technique, with rates of 87.5% vs. 53.3%, respectively [11]. These two high-pressure irrigation techniques, when carried out two to three times during one procedure with 20 mL irrigation fluid, can result in unfavorable outcomes if irrigation fluids are not allowed to discharge from the nasal cavity, following passage through the Eustachian tube [11,12]. Additionally, though rare, high-pressure irrigation can lead to critical complications, as it may induce severe destruction of the tympanic bulla and cause traumatic inflammation within the inner ear [11,12]. In domestic ruminants, conservative therapy can commonly lead to therapeutic success for acute otitis media [10]. However, surgical treatment is sometimes required in animals with chronic or end-stage otitis media, which is difficult to treat using these conservative treatments [15].

In previous cases, affected calves have been treated with bulla osteotomy using lateral and ventral approaches [6,7,8,16,17,18]. Two types of lateral bulla osteotomy (LBO) have been reported. One LBO procedure was total ear canal ablation followed by curetting of the neighboring lateral surface of the tympanic bulla, this being the common method performed on sick ruminants such as llama, alpaca, camelid, buffalo, bison, and bongo [10,13,19,20,21,22]. The other previously documented LBO procedure was conducted by the following steps: (1) an incision was made in the lower area of the ear; (2) the surgical opening was extended deeply by blunt dissection, allowing the exposure of the lateral wall of the tympanic bulla in the deeper site of the stylohyoid bone; and (3) perforation of the lateral wall of the tympanic bulla was made via the surgical opening [17,18]. However, these methods are so invasive that they generate a high probability of postoperative complications such as neurological disturbances, e.g., issues with the facial nerve, hypoglossal nerve, and vestibular nerve [10,15,20]. Thus, care should be taken to avoid injury of the facial arteries and nerves located on the surgical courses while performing both types of LBOs [15,16,17]. Additionally, LBOs with total ear canal ablation are applicable to ruminants with concurrent involvements of otitis externa and otitis media/interna, allowing the removals of both lesions [10,15]. Total ear canal ablation has no therapeutic effect for the bovine cases commonly presenting with otitis media (without otitis externa), caused by the spread of the pathogen via the Eustachian tube from the area of the nasopharynx [8,10,23].

Some previous reports have discussed ventral bulla osteotomy (VBO) as a minor surgical option in bovine otitis media, including detailed methodological explanations [6,7]. Regardless of whether LBO or VBO is chosen as a surgical intervention, surgical planning should be supported by preoperative evaluations of the affected tympanic bulla; such planning can determine whether there is unilateral or bilateral involvement, which side is affected in unilateral cases, and the degree of extension and destruction. These pathological changes in the affected tympanic bulla can be effectively detected using imaging examinations, supporting therapeutic decisions such as whether surgical intervention is required, the selection of suitable surgical techniques for each case, and diagnostic uses [2,3,4,5,6,10,11,15,17,19,20,21,24]. In terms of the convenient imaging modalities used in bovine practice, many previous reports have described the clinical applicability of radiography, computed tomography (CT), or ultrasonography [2,3,4,5,6,10,11,15,17,19,20,21,24]. This study aims to explain the role of CT evaluation prior to performing VBO in three calves with unilateral otitis media and discuss the applicability of CT. Additionally, this study includes the technical explanation of our VBO procedures, applied through a different surgical route to those carried out in previous bovine practice procedures [6,7] but one which is identical to the common VBO procedure in small-animal practice [23,25]. This technical explanation is followed by a discussion of CTs applicability and the problematic points of the present VBO procedure in comparison to the previous procedure, based on satisfactory or unsatisfactory outcomes.

## 2. Materials and Methods

A slip-ring CT scanner (Pronto SE, Hitachi Co. Ltd., Tokyo, Japan) was used in this study. A common therapeutic protocol for all three cases, including preoperative CT examination, and subsequent treatment with VBOs, was carried out while under general anesthesia. Anesthesia was composed of applying 2–3% isoflurane (Isoflo, DS Pharma Animal Health Co. Ltd., Osaka, Japan) via an endotracheal tube, inserted after sedation with an intravenous injection of xylazine hydrochloride (0.2 mg/kg, IV; Selactar 2%, Bayer Yakuhin Ltd., Osaka, Japan). Scanning was performed with the animal positioned in the dorsal recumbency on the examination table, with X-ray conditions of 100 kVp and 100 mA. In all three calves, CT was used to gather the tympanic bulla measurements of both ears. Each left and right tympanic bulla was measured for its dorsoventral height and mediolateral width on the transverse section, and the craniocaudal length on the sagittal section of the skull CT [5]. Additionally, the transverse CT section was used to measure the maximum thickness of the ventral wall of the tympanic bulla [4]. The values in these three animals were compared with those derived in the same manner from skull CTs in five Japanese Black calves (aged 30–60 days), and five Holstein calves (aged 67–140 days), which were examined during diagnosis of congenital deformities in the brain, eye and cervical vertebra, in addition to the diagnosis of pneumonia, arthritis, and a mandibular fracture.

## 3. Case Presentation

### 3.1. Case 1

A 4-month-old male Holstein calf with a body weight of approximately 100 kg had presented with a head tilt toward the right and right ear droop, which had been evident for the past month. The case presented normal physical performance in terms of appetite, drinking, and activity. Clinical signs did not respond to continuous administration of a penicillin and streptomycin combination solution (8000 U/kg and 10 mg/kg, SC, q24, 7 days; Mycillin, Meiji Seika Pharma Co., Ltd., Tokyo, Japan). On admission, the severity of the neurological findings was scored as 2, according to Bertone’s criteria [9].

On admission, CT confirmed an accumulation of hyperattenuating contents into the entire cavity of the right tympanic bulla. On the transverse section, the wall structures were entirely hypoattenuating, and thick and rounded in shape due to being extended medially and ventrally (Figure 1a). On the sagittal section demonstrating the right tympanic bulla, the hyperattenuating contents were included in the oblong tympanic cavity due to being extended cranially and ventrally (Figure 2a). No abnormality was evident in the left tympanic bulla. The height, width, and length of the right tympanic bulla were 33.0, 27.2, and 29.2 mm, respectively, and tended to be large compared with those in the opposite left tympanic bulla (24.8, 25.7, and 24.2 mm, respectively) and the measurements (mean ± standard deviation) from five Holstein calves (25.9 ± 4.0, 25.9 ± 2.2, and 24.0 ± 3.3 mm, respectively) (Table 1). Compared with the maximum thickness of the ventral walls in the ears of five Holstein calves (1.9 ± 0.4 mm), the values increased severely in the right affected tympanic bulla (10.9 mm), and slightly in the left tympanic bulla (3.4 mm). Based on quantitative evaluation, the extension in the ventral wall contributed to the larger height of the right tympanic bulla.

The animal was positioned in dorsal recumbency with the neck extended via a cushion beneath it. A 15 cm incision was made longitudinally in the right paramedian cervical skin caudally to the medial level of the angle of the mandible (Figure 3a). Through blunt dissection of the subcutaneous tissues, the sternothyroid and omohyoid muscles were identified as running in parallel and along the longitudinal incision (Figure 3b). Blunt separation between the sternothyroid and omohyoid muscles confirmed the right external carotid artery running along the trachea palpated at the medial aspect of the incision, allowing the observation of the right vagus nerve running in parallel with it. Two Gelpi retractors were used in the cranial and caudal aspects of the surgical opening: one hooked tip of the blade was applied to the lateral area including the omohyoid muscle and the right jugular vein; another was carefully applied to the medial area including the sternothyroid muscle, the right external carotid artery, the right vagus nerve, and the trachea (Figure 3c). The bone structure of the skull was exposed at the deepest area of the surgical opening, which was extended gradually as the handles were carefully closed and held by a ratchet in the Gelpi retractors. In just over 1 h, the ventral surface of the tympanic bulla could be identified as a rounded, irregular structure. The ventral surface of the tympanic bulla could be perforated manually with a hand chuck (Figure 3d), followed by an extension of the perforation with a bone rongeur. The caseous pus material presenting into the tympanic cavity was removed with a curette. Subsequently, the tympanic cavity was lavaged with warm saline solution before being suctioned. A drainage tube was not placed into the tympanic cavity. The sternothyroid and omohyoid muscles were interruptedly sutured with absorbable sutures to close the space between them. The skin incision was subsequently sutured using a nylon suture material. The total time required for the VBO was 2.5 h. The CTs carried out soon after surgery revealed a lack of structure in the entire caudal wall of the tympanic cavity, which was filled with irrigation fluid and air (Figure 2b,c). *Pasteurella multocida* was isolated in the irrigation fluid obtained from the tympanic cavity during surgery. A prednisolone solution (1 mg/kg, SC; Kyoritsu Seiyaku, Tokyo, Japan) and an oxytetracycline hydrochloride solution (5 mg/kg, SC; OTC 50% KS, Kyoritsu Seiyaku, Tokyo, Japan) were administrated intraoperatively. Postoperative care was performed by a seven-day administration of an oxytetracycline hydrochloride solution (5 mg/kg, SC, q24). The animal showed a quick improvement of clinical signs, such as the disappeared torticollis, and could develop normally.

### 3.2. Case 2

A 1-month-old male Japanese Black calf with a body weight of 35 kg suddenly presented a head tilt toward the right, right ear droop, and bilateral horizontal nystagmus. Severe depression was also evident, together with declined physical performance. The calf’s difficulty in standing was suspected to be associated with vestibular ataxia. On admission, because of the quick development of the clinical signs, the severity of the neurological findings was scored 7 [9]. On the preoperative CT, the hyperattenuating contents were seen in the entire right tympanic bulla, in which the wall structures were not thickened and extended (Figure 1b). No difference between the left and right tympanic bulla was found in the height (17.4 mm vs. 15.7 mm), the width (28.5 mm vs. 27.2 mm), the length (26.1 mm vs. 23.0 mm), or the thickness (1.6 mm vs. 2.4 mm), respectively (Table 1). These four values were almost similar to the measurements from the five Japanese Black calves (in order, 21.0 ± 2.0, 26.2 ± 2.6, 22.6 ± 2.4, and 1.9 ± 0.3 mm, respectively).

The surgery was performed using the same procedure as in Case 1. A 10 cm longitudinal incision was made in the tight paramedian cervical skin. The rounded ventral surface of the tympanic bulla was identified soon after the exposure of the ventral surface of the skull, using two Gelpi retractors on the extended surgical opening (Figure 4a). The ventral surface of the tympanic bulla was perforated manually with a bone rongeur (Figure 4b). Mucoid, cream-colored pus was removed with a curette (Figure 4c,d). After careful irrigation, a drainage tube was placed into the tympanic cavity. The tube was passed through the interruptedly sutured space between the sternothyroid and omohyoid muscles and the skin, and finally secured to the skin with Chinese finger trap suturing (Figure 4e). The total time required for the VBO was 65 min. *Mycoplasma bovis* and *Streptococcus suis* were isolated. For three postoperative days, the drainage tube was used for irrigation flushing with warm saline solution and was removed on the 3rd postoperative day. Medication consisted of an intraoperative administration of a prednisolone solution (1 mg/kg, SC) and an oxytetracycline hydrochloride solution (5 mg/kg, SC), followed by the postoperative administration of an oxytetracycline hydrochloride solution (5 mg/kg, SC, q24, 7 days). However, the clinical signs did not entirely improve in the animal, finally resulting in death on the 10th postoperative day.

### 3.3. Case 3

A 2-month-old male Japanese Black calf with a body weight of 45 kg had developed a head tilt toward the left and left ear droop roughly 10 days previously. Neither depression nor abnormal physical performance were evident. A therapeutic response could not be obtained from a single injection of tilmicosin phosphate solution (10 mg/kg, SC; Micotil300, Elanco Japan, Tokyo, Japan) followed by the administration of an oxytetracycline hydrochloride solution (5 mg/kg, IM, q24, 7 days). On admission, the severity of the neurological findings was scored 2 [9]. The preoperative CT revealed that the left tympanic bulla was slightly extended ventrally and included the entirety of the hyperattenuating contents into the cavity, despite the right tympanic bulla not being abnormal (Figure 1c). No difference between the left and right tympanic bulla was found in the width (25.8 mm vs. 27.7 mm, respectively) or the length (23.7 mm vs. 22.4 mm, respectively), although the height of the right tympanic bulla (21.2 mm) was slightly extended compared with that of the left tympanic bulla (18.4 mm) (Table 1). The thickness in the ventral wall of the left tympanic bulla (5.1 mm) was larger than the measurements from the five Japanese Black calves.

A 10 cm longitudinal incision in the left paramedian cervical skin confirmed that the sternothyroid and omohyoid muscles were running superficially in the surgical opening. Subsequently, two Gelpi retractors were carefully used to make a wider surgical opening, with one hooked tip of the blade applied to the medial area including the left external carotid artery and the left vagus nerve together with the sternothyroid muscle, and the other hooked tip applied to the lateral area of the surgical opening. At approximately 40 min after surgery, electrocardiography (ECG) revealed that tachycardia (>150 per min) and cardiac arrest started to be repeated periodically at intervals of 2–3 min when using an ECG recorder by an A-B lead for intraoperative monitoring (Appendix A). In one cardiac cycle, the cardiac waves comprised a missing P wave and of low-amplitude QRS and T segments in the tachycardia phase. This was followed by ECG content of only a T segment, as the magnitude of the QRS segments became gradually smaller and finally disappeared with a decreasing heart rate, and finally flattened into the cardiac arrest phase. After the cardiac arrest phase continued for several seconds, the heart rate suddenly increased, allowing the tachycardia phase to return. The periodic repeats of the abnormal cardiac cycles resulted in the complete cessation of the heartbeat. Before a perforation was made in the tympanic bulla, the animal died. Unfortunately, without the agreement of the owner, the animal could not be autopsied.

## 4. Discussion

Radiography is a routinely used imaging modality, providing distinctive findings of bovine otitis media, as represented by osseous changes in the affected tympanic bulla. These include thickened walls or osteolysis and increased soft tissue opacity, which is clearly distinguishable from the radiolucent lumen of the normally air-filled tympanic bulla [2,5,17,20]. Contrast radiography has previously confirmed the presence of a rupture in the tympanic membrane [19]. However, these radiographic characteristics can be difficult to demonstrate due to the visual obstacles of the peripheral skull’s structures and superimposition between both sides of the tympanic bulla [2,5]. The visual limitation can be resolved by observations of the lateral, ventrodorsal, and oblique radiographic views when taken from multiple directions of X-ray irradiation, despite the difficulty in proper positioning and requirement of sedation or anesthesia for the examined animals [5]. Compared with radiography, CT is considered the gold standard in antemortem examinations, as it enables the visualization of both tympanic bullae without their superimposition on the transverse section when scanned once in the same position of the anesthetized animals [5,20,24]. Ultrasonography also has a high diagnostic potential for bovine otitis media, with a sensitivity of 32–63% and specificity of 84–100% [24]. The ultrasonographic findings concerning bovine otitis media could be characterized mostly by the accumulations of the echogenic, heterogeneous contents within the lumens deeper than the hyperechoic walls, and a little by the irregular, thickened contours, and deformation of the walls [24]. However, ultrasonography may have less potential for surgical planning [11].

Bovine cases of otitis media commonly involve bilateral lesions within the tympanic bulla, even if presenting unilateral clinical signs; the prevalence of bilateral affection in animals with otitis media accounted for 60–70% [2,3,5]. In the three cases above, preoperative CT evaluations could provide positive evidence in judgments of surgical intervention using VBO, as the CT characteristics of otitis media were found unilaterally. The use of VBO led to a successful result in a previous calf with bilateral otitis media [6]. However, different surgical openings should be made for approaching each affected tympanic bulla; two VBO procedures can be carried out for both ears using the same dorsal recumbent position. Simply, this means the surgical time is doubled when performing VBO for bilateral otitis media. The level of surgical invasion can increase depending on the openings made on both sides of the neck. If LBO is the choice of surgical option for bilateral otitis media, a positional change is required so that the treated ear faces upwards while positioned in lateral recumbency. Thus, to reduce surgical invasion, unilateral involvements of otitis media should be identified preoperatively, followed by the detection of the lesion’s side, regardless of whether the condition is treated with LBO or VBO. In this respect, CT would be superior to radiography.

Radiography has previously been applied slightly obliquely to the skull of the anesthetized, recumbent animal, while keeping a metal landmark placed on its surface ventrally than the ear, before treatment with LBO [17]. Based on the relative location between the landmark and the affected tympanic bulla on the preoperative radiographic findings, accurately deciding whether to take the surgical route could contribute to minimizing operative invasion; a 5 cm skin incision could allow a pinpoint approach, compared with the common technique of LBO, in which a large, T-shaped skin incision is required for the wider surgical opening, as created by a >7 cm dorsoventral incision, followed by a >7 cm transverse incision made caudally from the dorsoventral cut line [17,18]. However, the LBO procedure has not been planned solely through using radiography, because of the difficulty in the evaluation of the degrees of extension and the changed shape of the affected tympanic bulla [17].

In bovines, chronic development of otitis media is a frequent cause of invasion of the peripheral structures [3]. Extension of the affected tympanic bulla is a typical osseous change associated with this disease’s chronicity, allowing mechanical loosening of the mobile function of the mandible within the temporomandibular joint [5]. Osteolytic changes that develop simultaneously within the extended tympanic bulla are the possible cause of infectious invasion to the meninges, together with elevated middle- and inner-ear pressure [3,5]. CT can provide sections from every direction and three-dimensional images can be built up through the reconstruction of consecutive scans, as well as the basic axial sections [26,27]. This function can allow the identification of the morphological relationship between the primary and secondary lesions, and the degree of the lesion’s extension, in diagnosis of various head diseases, inducing osseous abnormalities [26,27]. As demonstrated in this study, the measurements of the length and width in the middle-ear structures can confirm the severity of the extensive and destructive changes, providing useful evidence for surgical planning [4,5]. Additionally, in the evaluation of the CT values of osseous changes associated with otitis media with effusion, the reduced Hounsfield units display the increased severity of osteolytic changes, fragility, and the decreased degree of hardness of the walls of the tympanic bulla [28]. Based on these parameters, the preoperative, quantitative evaluation of CT is very helpful for determining whether to choose surgical therapy, selecting the preferable equipment, i.e., an electric or manual drill, and making preoperative decisions concerning the size and location of perforation made on the surface of the tympanic bulla [4,5]. In this study, despite the preoperative CT evaluations, a successful result from surgical intervention was obtained for one of the three cases; Case 1 showed a satisfactory outcome but Cases 2 and 3 died on the 10th postoperative day and during surgery, respectively. However, the unsatisfactory results in Case 2 and 3 can provide positive feedback in terms of surgical improvements in cases of bovine otitis media. The unsatisfactory outcome in Case 2 indicated that VBO might have less therapeutic efficacy for more severe cases of bovine otitis media. Case 2 presented with horizontal nystagmus, one of the critical vestibular signs, and showed a high neurological severity (score 7) [7,8,9,10]. A neurological score ≤ 5 is usually related to satisfactory results after two weeks of medical treatments for otitis media [9]. Severity of the clinical signs could also be related to the prognoses in the bovine cases treated with high-pressure irrigation, resulting in the different satisfactory outcomes in mild and severe cases (91.2% vs. 43.5%, respectively) [14]. Thus, clinical severity criteria should be identified to determine whether VBO is indicated while being evaluated by CTs.

In Case 3, the present VBO procedure induced abnormal cardiac rhythms, characterized as an alternation between tachycardia and cardiac arrest, resulting in intraoperative death. Despite never being reported in bovine practice, this phenomenon may be explained by a previous human case study [29]. The vagus nerve may be the efferent pathway responsible for bradycardia and cardiac arrest through the mechanical stimulation of the nerve and respiratory tract during endotracheal intubation and tracheal suction [29]. In the present VBO procedure, incisions were made caudally to the area of the thyroid cartilage. This is distinct from the previous procedure of VBO, in which an incision is made cranially to the area of the thyroid cartilage, in the anatomical space between the vertical ramus of the mandible and the paracondylar process of the occipital bone [6,7]. Thus, the difference between the previous and present procedures is in the main muscular layers presented on the surgical routes between the incision and the tympanic bulla. In the previous VBO procedure, these layers included the digastricus muscle (grouped in the muscles of mastication), and the hyoglossal and styloglossal muscles (grouped in the extrinsic lingual muscles), while in the present VBO procedure, these layers consisted of the sternothyroid muscle (grouped in the extrinsic laryngeal muscles) and the omohyoid muscle (grouped in the hyoid muscles) [6,7,30,31]. Additionally, the vagus nerve is present on this surgical route. During the present procedure of VBO, the trachea and the vagus nerve may have been stressed by the mechanical pressure generated by the Gelpi retractors. During VBOs in canine and feline cases, the use of self-retaining retractors is also unavoidable to maintain a good surgical view, and care must be taken to prevent injury of the nerves and vessels [23,25]. In this study, the length of the extended skin incisions depended on the body size of the treated animals; a 15 cm skin incision in Case 1 vs. a 10 cm skin incision in Cases 2 and 3. The extension of the skin incision may contribute to a reduction in the mechanical stress generated by the Gelpi retractors, although this increases surgical invasiveness.

To enhance the therapeutic effects obtained from VBO, it is recommended that a drain tube be placed into the tympanic cavity via the perforation made at the ventral wall of the tympanic bulla, to allow postoperative irrigation in bovine cases [7]. However, the drainage and irrigation effects rendered by the placing of a drainage tube have not been fully evaluated. Regardless of whether a drainage tube is used or not, the effusion and irrigation solution may naturally seep out via the perforation made ventrally, and retained within the surgical wounds. Based on the favorable result in Case 1, which was treated with VBO without the placing of a drainage tube, the applicability and necessity of using the drainage method should, therefore, be assessed, in terms of the technical development of VBO for the bovine cases.

In addition to preoperative and/or postoperative uses of CT in diagnosing, therapeutic planning, and evaluating the degree of surgical achievements, the intraoperative utilization of CT may represent an advanced technique to prevent intraoperative contingency and postoperative complications. Instead of the macroscopic operation via the surgical view obtained from the use of self-retaining retractors, the procedure of VBOs assisted by CT may contribute to a reduction in surgical time, >1 h in three present cases, because it is effective to perforate the tympanic bulla via a shortened skin incision. The imaging can provide therapeutic evidence to detect whether LBO or VBO is applicable for each case based on preoperative evaluations, such as the concurrent situation of otitis media/interna and otitis externa, degree of extension in the surrounding structures, and the osseous changes in the tympanic bulla, such as the degrees of expansion, thickening, and osteolysis. Based on the utility of CT in the three cases above, the severity of such osseous changes can be evaluated quantitatively by comparison with the normal ranges obtained from age-matched animals. Further utilization of CT for bovine otitis media is required to establish the age-based reference values of tympanic bulla sizes.

## Figures and Tables

**Figure 1 vetsci-09-00218-f001:**
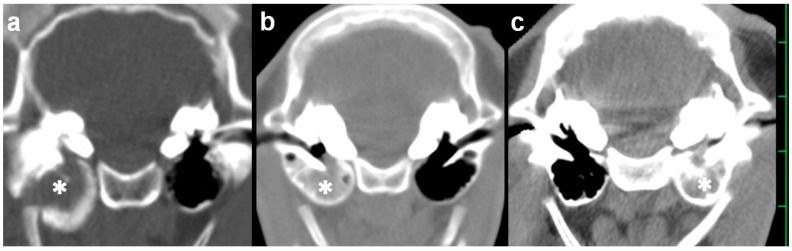
Transverse computed tomography of the skull examined preoperatively ((**a**) Case 1; (**b**) Case 2; and (**c**) Case 3). The hyperattenuating materials (asterisk) are seen within the tympanic cavity. The scale is 25 mm.

**Figure 2 vetsci-09-00218-f002:**
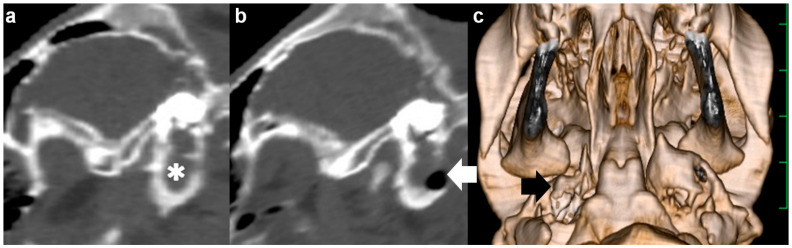
Sagittal computed tomography of the right tympanic bulla as examined preoperatively (**a**) and postoperatively (**b**) in Case 1. (**a**) The hyperattenuating materials (asterisk) are seen within the tympanic cavity. (**b**) A lack of bony structure (arrow) is seen in the entire caudal wall of the tympanic cavity. (**c**) Three-dimensional computed tomography showing the ventral surface of the skull in Case 1. A perforation (arrow) is made in the entire caudal wall of the tympanic cavity. The scale is 25 mm.

**Figure 3 vetsci-09-00218-f003:**
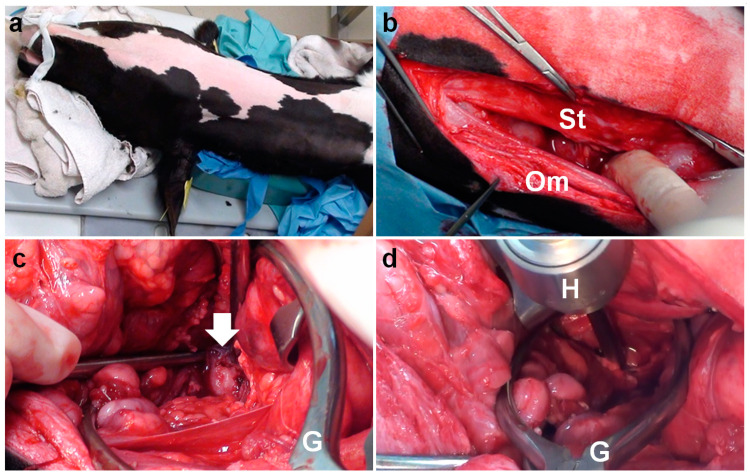
(**a**) Intraoperative photos of ventral bulla osteotomy in Case 1, positioned in dorsal recumbency with the neck extended. (**b**) The sternothyroid (St) and omohyoid (Om) muscles are separated under the area of a 15 cm incision. (**c**) The ventral surface of the tympanic bulla (arrow) is seen as a rounded, irregular structure at the deepest area of the surgical opening, extended using a Gelpi retractor (G). (**d**) A hand chuck (H) is used to perforate the tympanic bulla.

**Figure 4 vetsci-09-00218-f004:**
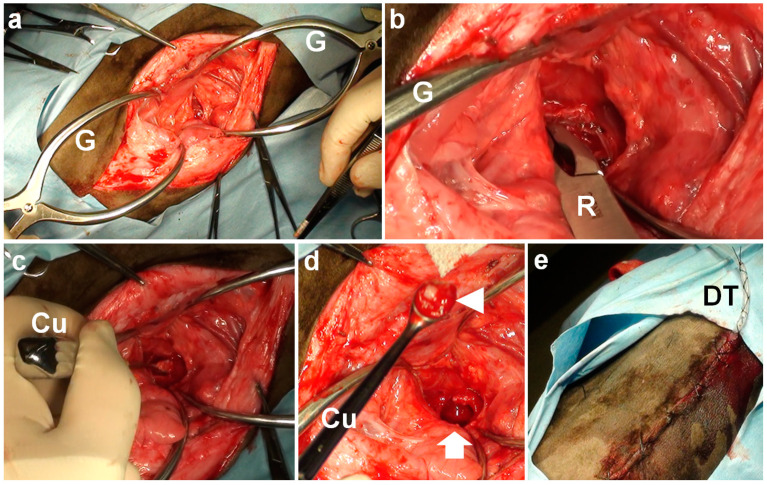
Intraoperative photos of ventral bulla osteotomy in Case 2. (**a**) Two Gelpi retractors (G) are used to extend the surgical opening. (**b**) A bone rongeur (R) is used to make a perforation in the tympanic bulla. (**c**) A curette (Cu) is inserted into the tympanic cavity. (**d**) The mucoid pus material (arrowhead) is subsequently removed from the perforated ventral surface of the tympanic bulla (arrow) using a curette (Cu). (**e**) A drainage tube (DT) is secured to the skin with Chinese finger trap suturing.

**Table 1 vetsci-09-00218-t001:** Computed tomographic measurements of tympanic bulla in three cases with otitis media and healthy calves.

Animals	Left/Right	Otitis Media	Dorsoventral Height	Mediolateral Width	Craniocaudal Length	Maximum Thickness of Ventral Wall
Case 1	Left	(−)	24.8	25.7	24.2	3.4
Case 1	Right	(+)	33.0	27.2	29.2	10.9
Case 2	Left	(−)	17.4	28.5	26.1	1.6
Case 2	Right	(+)	15.7	27.2	23.0	2.4
Case 3	Left	(+)	18.4	25.8	23.7	5.1
Case 3	Right	(−)	21.2	27.7	22.4	3.1
Japanese Black calves (number = 5)			21.0 ± 2.0	26.2 ± 2.6	22.6 ± 2.4	1.9 ± 0.3
Holstein calves (number = 5)			25.9 ± 4.0	25.9 ± 2.2	24.0 ± 3.3	1.9 ± 0.4

## Data Availability

The data presented in this study are available on request from the corresponding author.

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
