# Peer review of "Diagnostic and Therapeutic Evaluations of Computed Tomography in Three Calves with Unilateral Otitis Media Treated with Ventral Bulla Osteotomy"

_vetsci, 2022, doi:10.3390/vetsci9050218_

Round 1

Reviewer 1 Report

The authors must brush-up the English language of the text.

Author Response

Thank you for your kindly advise. The revised version is English-proofread by a English native speaker, belonging to English-proofreading company (Cambridge English Correction Service).

Reviewer 2 Report

This is a well written, interesting, case report that will contribute to the literature.

Author Response

Thank you for your kindly review’s work.

Reviewer 3 Report

The manuscript by Tsuka et al reports three cases of otitis media in calves, and also evaluated the applicability of computed tomography for the diagnostic and surgical intervention. It is interesting and helpful for the diagnostic use and supporting therapeutic decision. Here are some aspects that this reviewer concerns:

  1. Line 100, explain why choose the common VBO procedure in small-animal practice?
  2. Why the legend of Figure 1 and 2 are same? Please include the difference between these two figures in the legend.
  3. Line 180-182, do authors describe case 1 here? Why mention figures 2b, c that are for case 2 and 3?
  4. A drainage tube has been placed in case 2, but not case 1. What’s the criteria for placing the drainage tube?
  5. A 15-cm incision was made in case 1, while a 10-cm incision was made in case 3. What’s the size of incision in case 2? Please explain why the author made different size of the incision in each case. Could the shortened skin incision in case 2 and 3 be a reason for death?

Author Response

Comments and Suggestions for Authors

The manuscript by Tsuka et al reports three cases of otitis media in calves, and also evaluated the applicability of computed tomography for the diagnostic and surgical intervention. It is interesting and helpful for the diagnostic use and supporting therapeutic decision. Here are some aspects that this reviewer concerns:

Answer: Thank you for your kindly advise.

Question: Line 100, explain why choose the common VBO procedure in small-animal practice?

Answer: We are used to performing this procedure of VBO, because we are belonging to both small and large animal practice in Tottori University’s veterinary hospital. We are regularly performing this procedure of VBO for canine and feline cases with otitis media. Thus, in this study, this technique has been used for three bovine cases.

Question: Why the legend of Figure 1 and 2 are same? Please include the difference between these two figures in the legend.

Answer: Sorry, the legend of Figure 2 was mistaken completely. In the revised version, this legend is corrected.

Question: Line 180-182, do authors describe case 1 here? Why mention figures 2b, c that are for case 2 and 3?

Answer: Figure 2a,b,c shows the preoperative and postoperative CT images. In the revised version, the legend of Figure 2 is corrected.

Question: A drainage tube has been placed in case 2, but not case 1. What’s the criteria for placing the drainage tube?

Answer: In terms of the chronological order of three cases, we have performed the therapy for Case 2, Case 1 and Case 3, in order. As we can explain the story line of this manuscript, this manuscript was made by the order; Case 1 (favorable outcome), Case 2 (unfavorable outcome) and Case 3 (unfavorable outcome). We could experience that the drainage effect obtained from a drainage tube in Case 2 has been lower than we expected. Thus, in Case 1 having the larger body size than Case 2, we decided poor therapeutic effect of a drainage tube for Case 1. In the revised version, new sentences are added in lines 378-387.

Question: A 15-cm incision was made in case 1, while a 10-cm incision was made in case 3. What’s the size of incision in case 2?

Answer: In Case 2, a 10-cm longitudinal incision was made. In the revised version, the new sentence is added in lines 211-212.

Question: Please explain why the author made different size of the incision in each case. Could the shortened skin incision in case 2 and 3 be a reason for death?

Answer; In terms of the extension of skin incision, new sentences are added in the lines 373-377 in the revised version.

In the revised version, the corrected parts are highlighted by the yellow boxes.

This manuscript is a resubmission of an earlier submission. The following is a list of the peer review reports and author responses from that submission.